# Pharmacokinetic Outcomes of the Interactions of Antiretroviral Agents with Food and Supplements: A Systematic Review and Meta-Analysis

**DOI:** 10.3390/nu14030520

**Published:** 2022-01-25

**Authors:** Tippawan Siritientong, Daylia Thet, Janthima Methaneethorn, Nattawut Leelakanok

**Affiliations:** 1Department of Food and Pharmaceutical Chemistry, Faculty of Pharmaceutical Sciences, Chulalongkorn University, Bangkok 10330, Thailand; daliathet@gmail.com; 2Center of Excellence in Burn and Wound Care, Chulalongkorn University, Bangkok 10330, Thailand; 3Pharmacokinetic Research Unit, Department of Pharmacy Practice, Faculty of Pharmaceutical Sciences, Naresuan University, Phitsanulok 65000, Thailand; janthima.methaneethorn@gmail.com; 4Center of Excellence for Environmental Health and Toxicology, Naresuan University, Phitsanulok 65000, Thailand; 5Department of Clinical Pharmacy, Faculty of Pharmaceutical Sciences, Burapha University, Chonburi 20131, Thailand; nattawut.le@go.buu.ac.th

**Keywords:** food–drug interactions, nutrients, pharmacokinetics, HIV, AIDS

## Abstract

Because pharmacokinetic changes in antiretroviral drugs (ARV), due to their concurrent administration with food or nutritional products, have become a clinical challenge, it is necessary to monitor the therapeutic efficacy of ARV in people living with the human immunodeficiency virus (PLWH). A systematic review and meta-analysis were conducted to clarify the pharmacokinetic outcomes of the interaction between supplements such as food, dietary supplements, and nutrients, and ARV. Twenty-four articles in both healthy subjects and PLWH were included in the qualitative analysis, of which five studies were included in the meta-analysis. Food–drug coadministration significantly increased the time to reach maximum concentration (t_max_) (*p* < 0.00001) of ARV including abacavir, amprenavir, darunavir, emtricitabine, lamivudine, zidovudine, ritonavir, and tenofovir alafenamide. In addition, the increased maximum plasma concentration (C_max_) of ARV, such as darunavir, under fed conditions was observed. Area under the curve and terminal half-life were not significantly affected. Evaluating the pharmacokinetic aspects, it is vital to clinically investigate ARV and particular supplement interaction in PLWH. Educating patients about any potential interactions would be one of the effective recommendations during this HIV epidemic.

## 1. Introduction

The human immunodeficiency virus (HIV) infection is one of the major health challenges around the world. Approximately 33 million people have died from HIV/acquired immunodeficiency syndrome (AIDS)-related illness since the start of the HIV epidemic worldwide [1]. Globally, over 37 million people were living with HIV (PLWH) in 2020, of which more than 27 million were receiving HIV treatment or antiretroviral (ARV) therapy [2]. The commonly adopted ARV therapy includes the combined use of three or more ARVs from at least two different classes, such as nucleoside reverse transcriptase inhibitor (NRTI), non-nucleoside reverse transcriptase inhibitor (NNRTI), and protease inhibitor (PI). In adults and adolescents, tenofovir/lamivudine or tenofovir/emtricitabine is the preferred backbone, which is used in combination with a third drug, dolutegravir for post-exposure prophylaxis [3]. Due to the widespread use of effective ARV therapy and HIV care, about 66% of the overall infected population achieve viral suppression [2].

Dietary supplements including vitamins and minerals are widely used to boost the body’s defense system in many patients with nutrient deficiencies. European Food Safety Authority stated food supplements as sources of nutrients such as vitamins and minerals, which have nutritional or physiological value for the regulation of nutritional deficiencies [4]. In PLWH, inadequate nutrient intake becomes a clinical concern that could potentiate treatment failure. A previous report found that PLWH did not achieve the dietary recommendations of energy and micronutrients, especially zinc and iron [5,6]. To meet the required level of protein and other nutrients, PLWH use a variety of supplements in addition to their normal daily treatment therapy for HIV [7,8]. The use of complementary medications may be quite popular since the products are easily accessible over the counter. The lifetime use of complementary and alternative medicine was 30–90% prevalent in PLWH, commonly using vitamins, minerals, and other over-the-counter supplements [9].

In susceptible patients, even substantially small effects of food–drug interaction may result in therapeutic changes for some drugs with a narrow therapeutic index [10]. Likewise, food–drug interaction may influence the therapeutic efficacy of the drug by changing pharmacokinetic processes such as absorption, distribution, metabolism, and excretion, or pharmacodynamic physiological effects of the drug [11]. Given the different mechanisms of interactions between ARV and nutrients, it is critical to monitor the resulting implications that may positively or negatively affect the therapeutic outcomes. Due to the effect on pharmacokinetic parameters such as maximum plasma concentration (C_max_), area under the curve (AUC), time to reach maximum concentration (t_max_), and terminal half-life (t_1/2_), food considerations are essential in the treatment of HIV. The effect of food on the absorption of ARV was well noted following the reduction in plasma concentration of indinavir [12]. Food containing high-fat contents is likely to reduce the rate and extent of the absorption of oral drugs by delaying gastric emptying, whereas some meals with high-protein contents would increase the extent of oral drug absorption by stimulating intestinal transporters and enzyme activity [13]. However, the effects of food on particular oral drugs may not always be of clinical importance as some interactions may not occur in all patients. Although many ARVs can be taken with food to optimize their absorption, the concurrent administration with food might result in a decreased rate of absorption, longer t_max_, and declined C_max_ of some ARV such as zidovudine and lamivudine; although, they were not clinically significant [14]. Likewise, the subtherapeutic levels of raltegravir due to concomitant use of calcium supplement in an HIV-infected patient were reported [15]. Unpredictable and variable drug concentrations are major problems that lead to treatment failure or adverse reactions. The possible interaction of concurrent food and ARV has become a clinical challenge in PLWH. This systematic review and meta-analysis aims to investigate the effect of food, dietary supplements, or nutrients on pharmacokinetic outcomes of ARV by comparing the pharmacokinetic parameters in either PLWH or healthy people with and without supplements. The important role of food in the ARV era is not much known in clinical settings. The extent of changes in plasma concentration–time profiles of ARV during fed and fasted conditions can evaluate the potential interaction.

## 2. Materials and Methods

### 2.1. Literature Search

According to the Preferred Reporting Items for Systematic Reviews and Meta-Analyses (PRISMA) guidelines, a systematic literature search was conducted [16]. Four databases including Cochrane Library, ScienceDirect, Scopus, and PubMed were searched from the date of their inception to June 2021 to identify studies that explored supplements and ARV interactions. The search terms such as (food supplement OR dietary supplement OR nutrient supplement) AND (antiretroviral therapy OR antiretroviral drug OR non-nucleoside reverse transcriptase inhibitor OR nucleoside reverse transcriptase inhibitor OR protease inhibitor) AND (area under curve OR cytochrome P450 OR plasma concentration) were used. No filters or restrictions were applied during the search. The full search strategies are mentioned in Appendix A. The full protocol of this systematic review and meta-analysis was not registered.

### 2.2. Selection Criteria

The primary outcomes that we aimed to collect were AUC, C_max_, t_max_, and t_1/2_ which are parameters used for primarily assessed the food effect on drug pharmacokinetics [17]. All articles of any language, year published, and country that reported nutrient or food supplement and ARV interactions were included to retrieve relevant studies. Only studies that reported pharmacokinetic outcomes such as AUC, C_max_, t_max_, and t_1/2_ were included in the meta-analysis. Review articles, book chapters, conference abstracts, posters, in vitro studies, and animal studies were excluded. The search results were exported to a citation manager (Endnote 20.1., Clarivate Analytics, New York, USA). Titles and abstracts were thoroughly screened, and eligible studies were independently selected by the first two authors for inclusion in this systematic review. Specific characteristics for inclusion were studies of adult healthy people or PLWH on ARV, which discussed changes in ARV levels, concerned adverse events, or treatment failure directly resulting from the food–drug interaction (Phase I–IV clinical trials). Disagreements between the first two authors were thoroughly resolved by consensus, and further by discussing among four authors for final inclusion. The PRISMA diagram for the complete literature search is shown in Figure 1.

### 2.3. Data Extraction

After full-text articles were screened for inclusion in the systematic review, data extraction was performed. Information such as author name, year of publication, year of study, study design, study setting, characteristics of participants, types of ARV, types of supplements, and pharmacokinetic outcomes were extracted by the first author and the data entry was judged by co-authors. Discrepancies were discussed among all authors. Any disagreement was resolved by consensus.

### 2.4. Quality Assessment

The risk of bias of the included studies was assessed using the different validated tools including the risk of bias assessment tools for primary and secondary medical studies [18], ROBINS-I for non-randomized studies [19], Cochrane Collaboration’s tool for randomized trials [20], a modified Newcastle–Ottawa Quality Assessment Scale for cross-sectional studies [21], a modified tool for case report described by Murad et al. [22], and Cochrane risk of bias for crossover studies [23]. The quality assessment of included studies is mentioned in Appendix A.

### 2.5. Statistical analysis

Review Manager (RevMan version 5.4.1: The Nordic Cochrane Center, Copenhagen, Denmark) was used for conducting meta-analysis. Random effect models with the inverse variance method were used to calculate the weights of the studies. Publication bias was assessed by visually inspecting the funnel plot. The heterogeneity among studies was assessed by I^2^ statistics. The I^2^ of <25% was assumed as negligible heterogeneity, whereas that of >75% was high heterogeneity [20].

## 3. Results

A total of 3038 articles were retrieved. After duplicates and irrelevant studies were removed by reviewing titles and abstracts, 28 articles were selected for potential inclusion. The number of records with reasons for exclusion is summarized in Figure 1. Records that possibly met the inclusion criteria were assessed for eligibility for systematic review. Finally, twenty-four articles were included in the systematic review. Thirteen articles of them were good quality and eleven articles were fair (Appendix A). In general, the heterogeneity was high, and the 95% confidence intervals of the pooled data were close to the most weighted and included studies whose intervals were narrow. The summary description of the included studies is characterized in Table 1.

### 3.1. Studies Included in the Qualitative Analysis

Two-hundred and seventy-nine healthy subjects, and five-hundred and six PLWH who participated in twenty-four studies were included in the qualitative analysis (Table 1). The participants include various ethnic and racial groups such as people of European, African, and Asian descent. There were 3 randomized controlled trials, 13 cross-over studies, 2 longitudinal studies, 4 pharmacokinetic studies, 1case report, and 1cross-sectional study. Among the included studies, three studies reported the effects of grapefruit juice or Seville orange juice on ARV pharmacokinetics [24,26,30]. While nine studies investigated the effects of food or nutrient supplement on ARV pharmacokinetics [12,14,25,27,32,38,42,43,44]. The rest of the studies explored the pharmacokinetic interactions between ARV and vitamins or minerals [15,29,34,36,37,40], and dietary supplements [28,31,33,35,39,41]. Most studies showed non-significant changes in ARV pharmacokinetic parameters when co-administered with meals. However, there was a significant decrease in the extent of absorption of indinavir with liquid meals reported by Carver et al. [12]; protein, carbohydrate, fat, and viscosity meal treatments reduced indinavir AUC_0_–_∞_ by 68%, 45%, 34%, and 30%, respectively, while these meals decreased C_max_ by 74%, 59%, 46%, and 36%, respectively. Regarding the meal compositions, fat meals resulted in the highest variability on the plasma indinavir levels. The consumption of garlic supplement decreased AUC and C_max_ of saquinavir by 51% and 54%, respectively [28], while other dietary supplements such as quercetin, Ginkgo extract, Ginseng extract, β-carotene did not significantly affect saquinavir, lopinavir, ritonavir, and nelfinavir pharmacokinetics [31,33,35,41]. Grapefruit juice administration increased the oral bioavailability of saquinavir [24] while its concurrent intake did not lead to any change in indinavir concentrations [26]. A pharmacokinetic study by Slain et al. [29] indicated that after a week of 1000 mg vitamin C supplementation, the steady-state indinavir plasma concentration was reduced by 20%. On the contrary, some studies have reported that the use of multivitamins or minerals did not lead to any pharmacokinetic variation in ARV such as dolutegravir and nelfinavir [34,36,37,40]. In a case report of an HIV-infected man receiving raltegravir, concomitant calcium administration resulted in virologic failure showing detectable plasma HIV-1 RNA levels [15]. Coadministration of protease inhibitors such as nelfinavir with β-carotene led to a significant increase not only in mean CD4%, but also in the CD4:CD8 ratio in PLWH in one study [41]. In addition, the study also found an increase in CD4 counts; although, it was not clinically significant.

### 3.2. Studies Included in the Meta-Analysis

A total of 122 participants from 5 studies were included in the meta-analysis. Studies by Moore et al. [14], Yuen et al. [25], Falcoz et al. [27], Sekar et al. [32], and Yamada et al. [38] reported the pharmacokinetic outcomes of the interaction between food and ARV were eligible for meta-analysis. In general, the heterogeneity was high, and the 95% confidence intervals of the pooled data were close to the most weighted and included studies whose intervals were narrow.

Figure 2 shows the meta-analysis of the pharmacokinetic effects of food on the AUC_inf_ of ARV. There were no significant changes in AUC_inf_ of ARV between fasted and fed conditions (mean difference -24.64, 95% CI -141.34to 92.07, *p* = 0.68), with a substantial heterogeneity of 75%.

The overall effect of food on C_max_ of NRTI group shows no clinical significance between fasted and fed conditions (mean difference −0.56, 95% CI -2.31 to 1.20, *p* = 0.53) in fixed effect models, while C_max_ of NRTI group shows a significant decrease during fed state (mean difference -204.70, 95% CI -337.14 to -72.26, *p* = 0.002) in random effect models with the heterogeneity value of 84%. Interestingly, there was a significant increase in the C_max_ of PI group during fed state (mean difference 845.51 (110.10-1580.92, *p* = 0.02) as shown in Figure 3. Figure 4 indicates the overall effect of food on C_24_, which was not significant (mean difference −0.13, 95% CI −3.08 to 2.82, *p* = 0.93).

The results of the subgroup meta-analysis of pharmacokinetic changes in t_max_ of ARV depending upon food or supplement administration are shown in Figure 5. The overall effect in NRTI group indicates a significantly higher t_max_ value under fed condition (mean difference 0.67, 95% CI 0.44-0.91, *p* < 0.00001), showing the heterogeneity of 63%. Similarly, we found a significant increase in t_max_ in fed state compared to fasted condition in PI receiving groups (mean difference 1.29, 95% CI 0.81-1.77, *p* < 0.00001) (Figure 5). For both NRTI and PI groups, none of these show any clinically significant changes in t_1/2_ (Figure 6). We observed no statistical heterogeneity on t_1/2_ of both NRTI and PI regimens.

### 3.3. Publication Bias

Publication bias was assessed by visual inspection of the funnel plot, which is shown in Figure 7. It was found that there were seven missing studies for t_max_. The distribution of the standard error was close to the peak of the funnel plot, which demonstrated that there may be systematic difference or publication bias between smaller and larger studies. Further statistical test for the funnel plot asymmetry was not performed since the forest plot that had the largest number of included studies included only five studies [45].

## 4. Discussion

This systematic review and meta-analysis mainly presented no significant pharmacokinetic changes in AUC, C_24_, and t_1/2_ of ARV when co-administered with food or dietary supplements. However, it can be highlighted that there was an increase in t_max_ of NRTI and PI under fed conditions. Of interest, the higher C_max_ in fed state was observed in participants receiving the PI regimen in this study. Among the articles included in this systematic review, most of the ARV’s pharmacokinetic parameters were not clinically altered by the concurrent administration of food and other supplements such as quercetin, Ginkgo extract, vitamin supplement, Ginseng, or β-carotene.

The different classes of ARV have varied pharmacokinetic metabolisms. PIs are extensively metabolized by the cytochrome P (CYP) 450 enzymes in the liver and small intestine [46]. Thus, ingestion of some foods or juices that have an inhibitory effect on CYP450 may increase concentrations of PIs. Kupferschmidt et al. [24] reported an increase in the bioavailability of saquinavir bioavailability after an intake of grapefruit juice, which is widely known as a CYP3A4 inhibitor [47]. In addition, a previous study reported an increase in AUC of atazanavir, lopinavir/ritonavir, nelfinavir, and saquinavir when ingested with food [48]. Under the fed condition, there was a long t_max_ of amprenavir from 1 to 4 h compared with the fasting state [27]. The t_max_ of darunavir under fasted conditions was approximately 1.5 h, whereas it was increased to 3-4 h after administration with food including standard breakfast, croissant with coffee, and high-fat breakfast. Similarly, the median t_max_ of 2 h of ritonavir without food extended to 4 or 5 h when food was administered concurrently [32]. Regarding the changes in plasma concentration of PI such as darunavir, Sekar et al. [32] reported the increased C_max_ by about 30% under fed condition compared with the fasted state. Another class of ARV, NNRTIs such as efavirenz and nevirapine are also metabolized by several liver CYP isoenzymes [49]. The administration of efavirenz with high fat or high caloric meal was associated with the increased mean AUC by 28% and mean C_max_ by 29%, respectively, compared to fasted conditions [50]. Most NRTIs are degraded by liver enzymes from the purine or pyrimidine nucleoside salvage pathway, depending upon the NRTI analogs [51]. Since they are not extensively metabolized by CYP450 [51], they have less interaction with food or other drugs. The delayed absorption of abacavir, lamivudine, and zidovudine was noted by Yuen et al. [25] an hour later in median t_max_ under fed conditions compared to fasted conditions. Similarly, Moore et al. [14] observed the slower t_max_ of lamivudine and zidovudine by 30 min and 45 min, respectively, after the administration of combined tablets with a high-fat meal. The mean t_max_ of emtricitabine and tenofovir alafenamide under standard breakfast was 2 h and 1 h, respectively, which was longer than those under fasted conditions [38,52]. The concurrent administration of integrase inhibitors (IIs) and polyvalent cations such as magnesium and calcium should be monitored since these cations may bind with ARV, which leads to decreased plasma levels. Although Buchanan et al. [37] reported no clinical significance, dolutegravir, according to its medication package insert, should be taken 2 h before or 6 h after administering a cation containing antacids or laxatives, sucralfate, oral iron supplements, oral calcium supplements, or buffered medications [36,53].

The impacts of vitamin and mineral supplements on ARVs were mostly investigated in IIs [15,34,36,37] and some PIs [29,40]. Co-administration of 1200 mg calcium carbonate and 324 mg ferrous fumarate with dolutegravir during fasted conditions decreased in AUC_0–∞_, C_max_, and C_24_ of dolutegravir by respective 37% to 39% and 54% to 57% due to chelation with polyvalent cations [36]. However, coadministration of these divalent cations and dolutegravir under fed conditions counteracted such the interaction [36]. Moreover, Roberts et al. [15] reported subtherapeutic raltegravir levels when co-administered with 1000 mg calcium carbonate. Nonetheless, information on the time interval between raltegravir and calcium intake or whether they were administered under fed or fasted conditions were not reported. Given that calcium supplement is necessary for the prevention of osteoporosis, especially in PLWH [54], future studies investigating the magnitude of such interaction between raltegravir and calcium supplement under fasted and fed conditions should be conducted. If the fed condition can counteract the interaction, calcium supplements may be co-administered with raltegravir. Concerning the impacts of vitamin and mineral supplements on PIs, while the divalent calcium ion did not significantly alter nelfinavir and its active metabolite M8 concentrations [40], co-administration with vitamin C significantly decreased steady-state indinavir concentrations [15]. This could be explained by the inductive property of vitamin C on CYP isoenzymes observed in animal studies [55,56,57,58,59]. However, another study reported no significant effect of vitamin C on CYP3A4, the primary enzyme used to metabolize indinavir activity [60]. Based on the contradictory effects of vitamin C on CYP3A4, further studies are needed to confirm the results.

Concerning the impacts of various dietary supplements, different classes of ARVs were investigated, mostly the PIs. Piscitelli et al. [28] reported that garlic supplement significantly decreased the extent of saquinavir absorption, as indicated by the decrease in AUC_0–8_, C_trough_, and C_max_. No definite underlying mechanism could be drawn from this study, but the authors proposed that it could be due to the induction of CYP450, or P-glycoprotein (P-gp) produced by long-term use of garlic. Nonetheless, evidence has shown that the effects of garlic on CYP3A4 are controversial as some in vitro studies reported that garlic had an inhibitory effect on CYP3A4 [61]. While other studies showed no significant effect of garlic on CYP3A4 [62,63]. Given that garlic supplements might have a negative effect on saquinavir exposure, garlic supplementation should be avoided in patients receiving treatment with saquinavir.

Quercetin, an inhibitor of various CYP450 including CYP3A4 as well as P-gp, could theoretically be used as a booster of saquinavir levels [64,65]. However, DiCenzo et al. [31] did not find significant effects of quercetin on saquinavir concentrations. The nonsignificant effect of quercetin could partially be explained by the intersubject and intrasubject variability. Given that only ten subjects were included in the study by DiCenzo et al. [31], further studies are required to confirm such results.

*Ginkgo biloba*, another widely used dietary supplement, has been shown to induce CYP3A activity [66,67]. However, Robertson et al. [33] reported no significant effects of *Ginkgo biloba* extract on lopinavir and ritonavir pharmacokinetics. These could be explained by the inhibitory effect of co-administered ritonavir on CYP3A. Nevertheless, the impacts of *Ginkgo biloba* extract on un-boosted protease inhibitors were not investigated in that study, and hence coadministration of *Ginkgo biloba* extract with un-boosted PI is not theoretically recommended. Similar to the effect of *Ginkgo biloba*, *Panax ginseng* has shown an inductive effect on CYP3A, which in turn may reduce PI concentrations [68]. However, the two-week administration of *Panax ginseng* did not alter lopinavir and ritonavir pharmacokinetics [35]. The same reason concerning the inhibitory effect of concurrently administered ritonavir could be applied here. Therefore, a similar recommendation on the use of *Panax ginseng* and un-boosted PI to that of *Ginkgo biloba* extract is proposed.

Inconclusive effects of β-carotene on CYP3A4 have been reported from in vitro studies, as one study found an inhibitory effect [69], while another reported an inductive property [70]. A clinical study investigating the impacts of β-carotene supplementation at the dose of 25,000 IU twice daily on nelfinavir and its active metabolite M8, which is metabolized by respective CYP2C19 and CYP3A4, indicated no clinically significant effect [41]. This implies that PLWH receiving ARVs that are substrates of CYP2C19 or CYP3A4 may be able to use β-carotene as a dietary supplement. However, it should be noted that these findings were based only on 15 HIV-infected subjects. Further prospective studies investigating the impact of β-carotene on other CYP2C19 and CYP3A4 substrates may be warranted.

This study has some limitations. The included studies did not report the component of fruit juices, i.e., whether the juice contained pulp or additives. This information may be crucial since these compositions might not be pharmaceutically inert. Since most studies were conducted with different ARV drugs and different supplements, only limited studies were included for meta-analysis. Only food (meal) and ARV pharmacokinetic interaction in healthy subjects were analyzed due to the limited number of studies with supplements and ARV pharmacokinetic interaction. Additionally, caution should be taken when the stratified results are interpreted as the minimal number of studies for subgroup analysis should be more than 10 [71]. Some studies from the qualitative analysis could not be included in the meta-analysis because of the diverse types of supplements among studies. The number of participants in each study was relatively small, and there may also have been interpatient or intrapatient variability. Furthermore, most studies were conducted in different regions with different study designs. Moreover, our systematic review may not be generalizable to all nutrient or food supplements as we did not focus on some products, e.g., probiotics or prebiotics, which may have effects on ARV pharmacokinetics. Of note, the heterogeneity across studies may also affect the results due to the potential effects of some factors such as gender, comorbidities, food compositions, dosage regimen, and duration of treatment.

This study found new information on the potential impact of supplement use on ARV pharmacokinetics. The study highlighted the decreased absorption of NRTIs such as abacavir, emtricitabine, and PIs such as ritonavir and darunavir after co-administration with food. The increased t_max_ and plasma concentration of some ARV such as darunavir due to ARV–food interaction can suggest nutrition monitoring in relation to HIV and ARV treatment. As a further matter, due to the diversity of supplements, further research that considers not only the variability of interactions but the likelihood of an individual patient to develop potential outcomes should be examined. Improving awareness of therapeutic monitoring followed by pharmacokinetic evaluation of specific interaction should be implemented because there is extensive use of supplements in the HIV population.

## 5. Conclusions

This study pointed out the delayed absorption of NRTIs such as abacavir, emtricitabine, lamivudine, zidovudine, tenofovir alafenamide, and PIs including amprenavir, darunavir, and ritonavir under fed conditions including low-fat and high-fat meals. The higher plasma concentration of PI such as darunavir and cobicistat was also observed in a fed state. Considering the effects of various products on drug-metabolizing enzymes, more evidence in pharmacokinetic fields is required. So far, it is still necessary to investigate the potential interactions of ARV and particular supplement or complementary products in PLWH. Further pharmacokinetic studies are highly recommended to explore the potential pharmacokinetic interaction between food and ARV that might affect therapeutic outcomes. Physicians and healthcare providers need to be aware of the potential interaction between prescribed ARV and any mineral supplements that may lead to virologic failure or delayed absorption due to chelation. Since even a small change in plasma drug level can affect therapeutic efficacy, PLWH on ARV should be educated or given information on supplement use.

## Figures and Tables

**Figure 1 nutrients-14-00520-f001:**
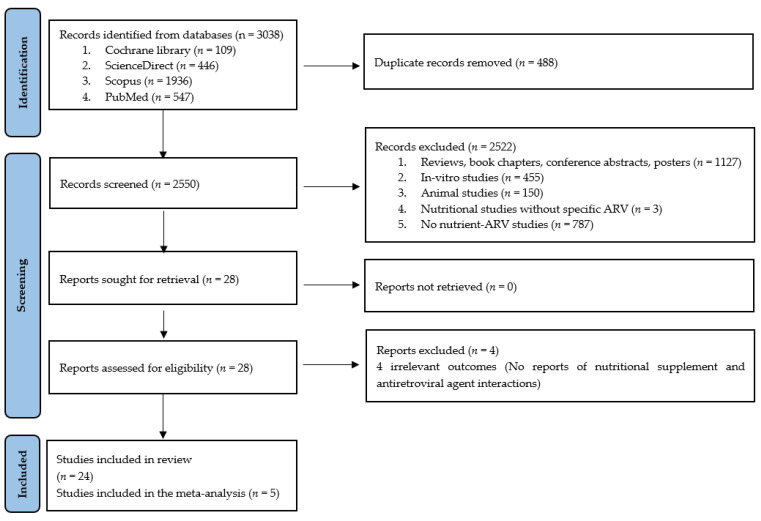
Flow diagram for selection and inclusion of the studies.

**Figure 2 nutrients-14-00520-f002:**
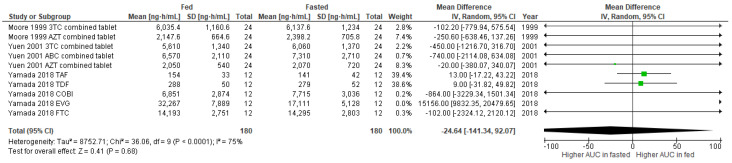
Forest plot showing the mean difference in AUC_inf_ of ARV under fasted and fed states (3TC, lamivudine; ABC, abacavir; AZT, zidovudine; COBI, cobicistat; EVG, elvitegravir; FTC, emtricitabine; TAF, tenofovir alafenamide; TDF, tenofovir).

**Figure 3 nutrients-14-00520-f003:**
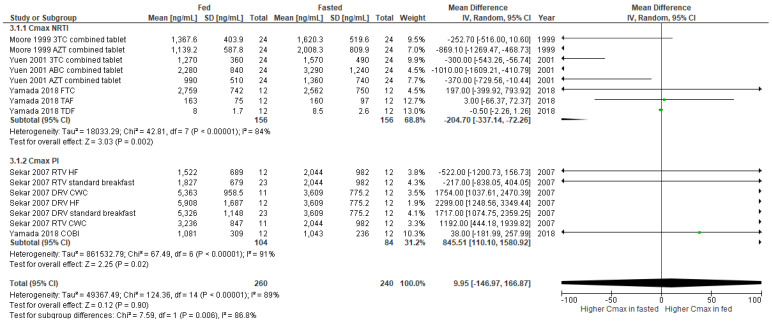
Forest plot showing the mean difference in C_max_ of ARV under fasted and fed states by ARV regimens (3TC, lamivudine; ABC, abacavir; AZT, zidovudine; COBI, cobicistat; CWC, croissant with coffee; DRV, darunavir; FTC, emtricitabine; HF, high fat; NRTI, nucleoside reverse transcriptase inhibitor; PI, protease inhibitor; RTV, ritonavir; TAF, tenofovir alafenamide; TDF, tenofovir).

**Figure 4 nutrients-14-00520-f004:**
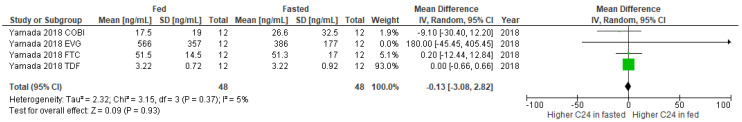
Forest plot showing the mean difference in C_24_ of ARV under fasted and fed states (COBI, cobicistat; EVG, elvitegravir; FTC, emtricitabine; TDF, tenofovir).

**Figure 5 nutrients-14-00520-f005:**
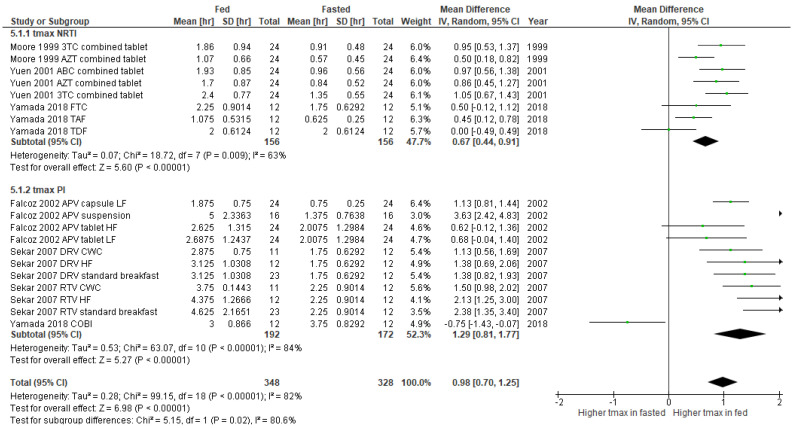
Forest plot showing the mean difference in t_max_ of ARV under fasted and fed states by ARV regimens (3TC, lamivudine; ABC, abacavir; APV, amprenavir; AZT, zidovudine; COBI, cobicistat; CWC, croissant with coffee; DRV, darunavir; FTC, emtricitabine; HF, high fat; LF, low fat; NRTI, nucleoside reverse transcriptase inhibitor; PI, protease inhibitor, RTV, ritonavir; TAF, tenofovir alafenamide; TDF, tenofovir).

**Figure 6 nutrients-14-00520-f006:**
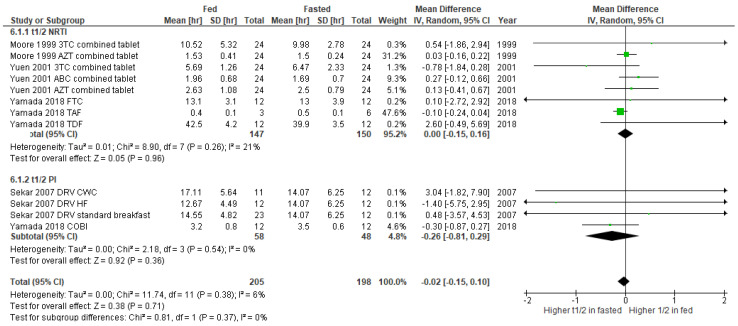
Forest plot showing the mean difference in t_1/2_ of ARV under fasted and fed states by ARV regimens (3TC, lamivudine; ABC, abacavir; AZT, zidovudine; COBI, cobicistat; CWC, croissant with coffee; DRV, darunavir; FTC, emtricitabine; HF, high fat; NRTI, nucleoside reverse transcriptase inhibitor; PI, protease inhibitor; TAF, tenofovir alafenamide; TDF, tenofovir).

**Figure 7 nutrients-14-00520-f007:**
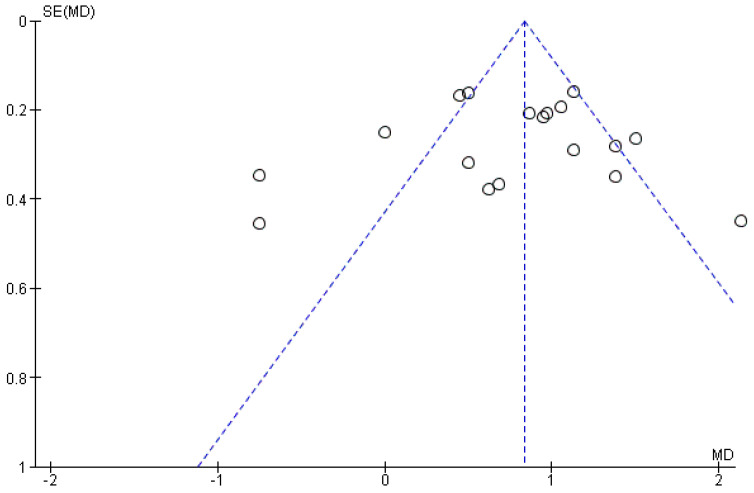
Funnel plot of the adjusted association between t_max_ and food (SE, standard error; MD, mean difference).

**Table 1 nutrients-14-00520-t001:** Description of the included studies.

Author, Year of Publication	Study Setting	Study Design	Study Participants	Food or Dietary Supplements	Antiretroviral Drugs	Major Outcomes
Kupferschmidt et al., 1998 [24]	Switzerland	Open crossover study	8 healthy male subjects, mean age 26 ± 2 years	400 mL grapefruit juice (Hitchcock, freshly prepared), taken 45 min and 15 min before intravenous or oral saquinavir	12 mg intravenous saquinavir or 600 mg oral saquinavir	- Grapefruit juice enhances the oral bioavailability of saquinavir without affecting its clearance. - AUC of saquinavir increased after pretreatment with grapefruit juice.
Moore et al., 1999 [14]	United States of America	Single-center, open-label, randomized, three-way crossover study	24 healthy subjects, mean age 26 ± 4.4 years	The standardized high-fat breakfast: 2 slices of toasted white bread with butter, 2 eggs fried in butter, 2 slices of bacon, 2 ounces of hash-browned potatoes, and 8 ounces of whole milk	150 mg lamivudine, 300 mg zidovudine, taken within 5 min after breakfast	- The extent of absorption of lamivudine and zidovudine from the combination tablet was not altered by administration with meals. - The absorption rate (t_max_ and C_max_) of lamivudine and zidovudine was reduced. - Since these changes were not considered clinically important, the drug formulation can be taken with or without food.
Yuen et al., 2001 [25]	United States of America	Single-center, open-label, randomized, three-way crossover study	24 healthy subjects, mean age 37.6 ± 9.6 years	The standardized high-fat breakfast: 2 slices of toasted white bread with butter, 2 eggs fried in butter, 2 slices of bacon, 2 ounces of hash-browned potatoes, and 8 ounces of whole milk	300 mg abacavir, 150 mg lamivudine, 300 mg zidovudine, taken within 5 min after breakfast	- The extent of absorption of abacavir, lamivudine, and zidovudine from the combination tablet was not altered by administration with meals. - The absorption rate (t_max_ and C_max_) of abacavir, lamivudine, and zidovudine were reduced. - Since these changes were not considered clinically important, the drug formulation may be taken with or without food.
Penzak et al., 2002 [26]	United States of America	Open-label, randomized, crossover study	13 healthy subjects, mean age 24 ± 1.9 years	8 ounces of Seville^®^ orange juice (prepared by squeezing fresh fruit) or grapefruit juice (prepared from frozen concentrate), taken together with indinavir	800 mg indinavir	- Coadministration of Seville^®^ orange juice and indinavir resulted in a statistically significant increase in indinavir t_max_ without altering other pharmacokinetic parameters. - Coadministration of grapefruit juice and indinavir did not significantly change indinavir pharmacokinetic parameters including AUC_0–5_, AUC_0–8_, C_min_, C_max_, t_1/2_, and oral clearance of indinavir.
Falcoz et al., 2002 [27]	Germany	Study A: Single-center, open-label, single-dose, randomized, five-way crossover study	16 healthy male subjects, age range 24–50 years	The standardized high-fat breakfast: 2 slices of toasted white bread with butter, 2 eggs fried in butter, 2 slices of bacon, 2 ounces of hash-browned potatoes, and a glass of whole milk	1656 mg GW433908A, 1728 mg GW433908G (taken within 5 min after breakfast), 2592 mg GW433908G, 1200 mg amprenavir	- The effect of food on GW433908G pharmacokinetic parameters (AUC_0–∞_, C_max_, t_max_, and oral bioavailability) was not statistically significant. - The therapeutic levels of amprenavir prodrug under fed conditions and fasting state were comparable.
		Study B: Open-label, single-dose, randomized, six-way crossover bioequivalence study	24 healthy male subjects, age range 19–48 years	- The standardized high-fat breakfast: 2 slices of toasted white bread with butter, 2 eggs fried in butter, 2 slices of bacon, 2 ounces of hash-browned potatoes, and a glass of whole milk - The low-fat meal: 30 g cornflakes, 100 g semi-skimmed milk, and 2 slices of toasted white bread with margarine and marmalade	1728 mg GW433908G (taken after low- or high-fat meal), 1200 mg amprenavir (taken after low-fat meal)	
Piscitelli et al., 2002 [28]	United States of America	Two-treatment, 3-period, single-sequence, longitudinal study	9 healthy subjects, mean age 38 ± 7.8 years	Garlic caplet	1200 mg saquinavir, taken together with garlic	In the presence of garlic, the mean saquinavir AUC_0–8_, C_trough_, and C_max_ were decreased.
Slain et al., 2005 [29]	United States of America	Prospective, open-label, longitudinal, two-period time series	7 healthy subjects, mean age 23.4 ± 1.6 years	1000 mg vitamin C	800 mg indinavir, taken at least 3 h separately from vitamin C	High doses of vitamin C can significantly reduce steady-state indinavir plasma concentrations by 20%.
Mouly et al., 2005 [30]	United States of America	Randomized, 2-phase, crossover study	20 healthy subjects, mean age 28 ± 9 years	Seville^®^ orange juice (prepared by squeezing fruit)	600 mg saquinavir	Seville^®^ orange juice delayed absorption of saquinavir (prolonged t_max_).
DiCenzo et al., 2006 [31]	New York, United States of America	Prospective pharmacokinetic analysis	10 healthy subjects, mean age 30.7 ± 9.4 years	500 mg quercetin with food (standardized light breakfast of 40 g of Cheerios^®^ cereal, 350 g of 2% milk, 43 g of toasted white bread, 9 g of margarine, and 4 g of sugar.)	1200 mg saquinavir, taken together with quercetin	Administration of quercetin did not influence plasma saquinavir concentration.
Sekar et al., 2007 [32]	Belgium	Open-label, 2-panel, randomized, 3-way crossover study	22 healthy subjects, median age 32 (2–50) years males, 49 (34–55) years females	- Standard breakfast: 4 slices of bread, 1 slice of ham, 1 slice of cheese, butter, jelly, and 2 cups of coffee/tea with milk and/or sugar - High-fat breakfast: 2 eggs fried in butter, 2 strips of bacon, 2 slices of white bread with butter, 1 croissant with 1 slice of cheese, and 240 mL of whole milk - Croissant with coffee	100 mg ritonavir bd on days 1 to 5, with a single 400 mg darunavir given on day 3 (darunavir/ritonavir), taken immediately after meal	Administration of darunavir/ritonavir in a fasting state resulted in a decrease in darunavir C_max_ and AUC_last_ of approximately 30% compared with administration after a standard meal.
Robertson et al., 2008 [33]	United States of America	Single-center, open-label	14 healthy subjects, mean age 29.5 (23–48) years	120 mg *Ginkgo biloba*	400 mg lopinavir/100 mg ritonavir, taken together with *Ginkgo biloba* extract	Neither lopinavir nor ritonavir pharmacokinetic parameters were significantly changed by 2 weeks of *Ginkgo biloba* extract coadministration.
Patel et al., 2011 [34]	-	Open-label, randomized, crossover study	16 healthy subjects, mean age 30.8 years	Multivitamin tablet (162 mg of elemental calcium and 100 mg of magnesium per tablet, in addition to iron, zinc, and copper)	50 mg S/GSK1349572, taken together with multivitamin tablet	Multivitamins did not significantly affect S/GSK1349572 pharmacokinetics. Therefore, they may be co-administered.
Calderõn et al., 2014 [35]	United States of America	Single sequence, open-label, single-center pharmacokinetic investigation	12 healthy subjects, median age 32 (23–42) years	500 mg *Panax ginseng*	400 mg lopinavir/100 mg ritonavir, taken together with *Panax ginseng*	Neither lopinavir nor ritonavir pharmacokinetic parameters were changed by two weeks of *Panax ginseng* administration.
Song et al., 2015 [36]	United States of America	Open-label, randomized, 2-cohort, 4-period crossover study	21 healthy subjects, mean age 33.2 years	1200 mg Calcium carbonate/324 mg Ferrous fumarate	50 mg dolutegravir, taken together with Calcium or iron supplements	- During mealtime, dolutegravir and calcium or iron supplements can be co-administered since the food increases the exposure. - Under fasted conditions, dolutegravir should be administered 2 h before or 6 h after administration of calcium or iron supplements, as there is a reduction in AUC_0–∞_, C_max_, and C_24_ of dolutegravir by chelation.
Buchanan et al., 2017 [37]	United States of America	Phase 1, single-center, randomized, open-label, 5-period crossover study	15 healthy subjects, mean age 39.8 ± 12.5 years	High-mineral content water (Contrex^®^: calcium 468 mg/L, magnesium 74.5 mg/L), Low-mineral content water (5% Contrex^®^ in purified water)	20 mg dolutegravir (dispersed in 12.5 mL of high- or low-mineral water	Dolutegravir pharmacokinetic parameters were unaffected by mineral contents in water.
Yamada et al., 2018 [38]	Japan	Open-label, randomized, single-dose, 3-treatment, 3-period, 3-sequence crossover study	12 healthy male subjects, mean age 32 ± 6.8 years	- Standard breakfast: 2 slices of bread with strawberry jam, 1 boiled egg, and 160 g of grape juice - Nutritional protein-rich drink (250 mL Ensure^®^ liquid)	150 mg elvitegravir, 150 mg cobicistat, 200 mg emtricitabine, 10 mg tenofovir alafenamide, taken within 5 min after breakfast	Food or a nutritional protein-rich drink did not affect the bioavailability of study drugs.
Yonemura et al., 2018 [39]	Japan	Open-label, randomized, single-dose, 3-treatment, 3-period, 3-sequence crossover study	12 healthy male subjects, mean age 30.7 ± 6.4 years	- Nutritional protein-rich drink (250 mL Ensure^®^ liquid) - 200 mL milk - 200 mL apple juice	Elvitegravir/cobicistat/emtricitabine/tenofovir alafenamide, taken within 5 min after supplements	- There were no differences in pharmacokinetic parameters of cobicistat. - Taking elvitegravir/cobicistat/emtricitabine/tenofovir alafenamide with milk could maintain therapeutic plasma concentrations of elvitegravir.
Carver et al., 1999 [12]	United States of America	Four-way crossover study	7 male PLWH, mean age 41 ± 18 years	Protein, carbohydrate, fat, HPMC separately provided in the form of low viscosity liquid meals, administered 15 min before indinavir	200 mg indinavir	- All meals decreased the extent (AUC_0–∞_ and C_max_) of indinavir absorption compared to fasted control.
Jensen-Fangel et al., 2003 [40]	Denmark	Open-label prospective randomized trial pilot study	15 PLWH on nelfinavir and reported chronic diarrhea, median age 51 (32–66) years	1350 mg calcium carbonate or calcium gluconate 1950/calcium carbonate 300 mg	1250 mg nelfinavir	There were no significant changes in plasma concentration (C_0h_ and C_3h_) of nelfinavir and its active metabolite M8.
Roberts et al., 2011 [15]	United States of America	Case report	An HIV-infected man	Calcium carbonate 1 g vitamin D3 400 IU (cholecalciferol)	400 mg raltegravir/200 mg emtricitabine + 300 mg tenofovir disoproxil fumarate	- After 10 months of starting raltegravir, the patient subsequently developed detectable HIV-1 RNA levels (7190 copies/mL) with documented resistance to raltegravir. - Calcium supplements may lead to subtherapeutic raltegravir levels due to binding to the divalent metal ion-chelating motif of raltegravir.
Sheehan et al., 2012 [41]	Canada	Multi-center, open-label, non-randomized steady-state pharmacokinetic study	11 PLWH receiving nelfinavir 1250 mg twice daily with at least two NRTIs for at least two weeks, mean age 45.5 ± 9.4 years	β-carotene 25,000 IU, taken together with nelfinavir	1250 mg nelfinavir	- β-carotene supplementation did not cause any clinically significant difference in the nelfinavir and M8 exposure. - Mean CD4% and CD4: CD8 ratio increased significantly.
Abdissa et al., 2015 [42]	Ethiopia	Double-blinded, randomized trial	282 ART-naïve PLWH, mean age 32.2 ± 8.0 years (LNS/w), 34.5 ± 10.3 years (LNS/s), 31.7±8.5 years (no LNS)	Lipid-based nutrient supplement containing whey (LNS/w) Lipid-based nutrient supplement containing soy (LNS/s)	600 mg efavirenz/200 mg nevirapine	- LNS intake was associated with lower plasma nevirapine trough concentrations, indicating possible drug–LNS interactions. - LNS did not affect EFV trough concentrations.
Munkombwe et al., 2016 [43]	Zambia	Randomized controlled trial	130 ART-naïve malnourished PLWH (BMI < 18kg/m^2^), mean age 35 ± 8 years (LNS), 38 ± 9 years (LNS-VM)	Lipid-based nutrient supplements (LNS) or LNS with vitamins and minerals (LNS-VM)	300 mg tenofovir disoproxil fumarate/200 mg emtricitabine/600 mg efavirenz	The LNS-VM regimen appeared to offer protection against phosphate and potassium loss during HIV/AIDS treatment.
Daskapan et al., 2017 [44]	The Netherlands	Cross-sectional study (short report)	60 PLWH receiving darunavir/ritonavir 800/100 mg od, median age 45 (20–66) years	Main meal/between-meal snack	800 mg darunavir/100 mg ritonavir	Concurrent food intake did not affect darunavir trough concentrations.

AUC: area under the curve; t_max_: time to reach maximum concentration; C0h: plasma concentration at 0 hr; C3h: plasma concentration at 3 h; Cmax: maximum plasma concentration; GW433908A: sodium salt of amprenavir prodrug; GW433908G: calcium salt of amprenavir prodrug; HPMC: hydroxypropylmethylcellulose; LNS: Lipid-based nutrient supplement; M8: an active metabolite of nelfinavir; PLWH: people living with HIV; S/GSK1349572: dolutegravir.

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
