# Peer review of "Pharmacokinetic Outcomes of the Interactions of Antiretroviral Agents with Food and Supplements: A Systematic Review and Meta-Analysis"

_nutrients, 2022, doi:10.3390/nu14030520_

Round 1

Reviewer 1 Report

This is a very interesting review on PLWH and currently significant with COVID-19 pandemic still ravaging most communities, but especially those in Low and Middle Income Countries(LMIC). These countries have been affected by supply chains that have upended medical and food supply chain. 

A majority of the articles however did not clearly indicate the spacing of the meals and the make up of the juices, that is whether they were pure or rich in pulp.

Further, it would have helped to indicate the CD4, CD8 counts and the viral load changes during the study period.

Additionally, articles should have included the intake of prebiotics and probiotics that are known to play a role in dysbiosis of the bacteria in the gut that may influence the ARV pharmacokinetics.  

Author Response

This is a very interesting review on PLWH and currently significant with COVID-19 pandemic still ravaging most communities, but especially those in Low and Middle Income Countries (LMIC). These countries have been affected by supply chains that have upended medical and food supply chain. 

Authors’ reply: We appreciate the reviewer’s remark and valuable suggestions. Responses to reviewer 1 comments are highlighted in cyan.

  1. A majority of the articles however did not clearly indicate the spacing of the meals and the makeup of the juices, that is whether they were pure or rich in pulp.

            - We mentioned the time taken between the administration of ARV and supplements or meals in Table 1. The studies of fruit juice did not mention the preparation or proportions of fruit juice in detail. We added a sentence to address this issue in discussion.

“The included studies fail to report the component of fruit juices i.e., whether the juice contains pulps or additives. This information may be crucial since these compositions might not be pharmaceutically inert.”

  1. Further, it would have helped to indicate the CD4, CD8 counts and the viral load changes during the study period.

            - Unfortunately, few studies indicated the changes (increase or decrease) in CD4, CD8 and HIV RNA levels after coadministration of ARV and supplements. The added highlighted sentences were mentioned under the result section “Studies included in qualitative analysis” in the manuscript. However, we added CD4, CD8, and HIV RNA levels, when the data are available, in Table 1.

  1. Additionally, articles should have included the intake of prebiotics and probiotics that are known to play a role in dysbiosis of the bacteria in the gut that may influence the ARV pharmacokinetics.

              - We did not include prebiotics and probiotics in the terms applied for the literature search. However, as a possibility of the gut microbiota on the ARV pharmacokinetics, we mentioned “we did not focus on some products e.g. probiotics or prebiotics which may have effects on ARV pharmacokinetics” in the Discussion. This would be one limitation of our study.

Reviewer 2 Report

Manuscript ID: nutrients-1534747

Comments and Suggestions for Authors

Major comments,

The authors conducted a systematic review and meta-analysis to clarify the pharmacokinetic outcomes of the interaction between supplements and antiretroviral agents (ARV). Basically, the pharmacokinetic outcomes were varied between the people were living with HIV (PLWH) and healthy subjects (HIV-negative). The authors should be clearly defined the primary outcome in the study design to stratify the two groups (PLWH and healthy subjects) and the classification of ARV. Hence, the authors need to carefully present the data in PLWH on ARV. Therefore, I suggest the authors perform the subgroup analysis. For primary outcomes, the pooled proportions between the PLWH and healthy subjects, as well as ARV were calculated, the extremities sample size was observed between the two groups. And the test heterogeneity was obviously found in Figures 2, 3, and 5. The authors should well discuss in the main text and in the limitation of the study. Given all the currently available protease inhibitors (PIs) are metabolised by the cytochrome P450 (CYP) enzyme system. All are inhibitors of CYP3A4, ranging from weak inhibition for saquinavir to very potent inhibition for ritonavir. (Malaty LI, Kuper JJ. Drug interactions of HIV protease inhibitors. Drug Saf. 1999 Feb;20(2):147-69.) Nucleoside reverse transcriptase inhibitors (NRTIs) do not undergo hepatic transformation through the CYP450. NRTIs would be less have the interaction with supplements. Non-nucleoside reverse transcriptase inhibitors (NNRTIs) are metabolized in the liver by CYP3A isoenzymes. Their pharmacokinetic outcomes were varied and complex. Also, the main concern regarding interactions with integrase strand transfer inhibitors (INSTIs) is the potential for decreased absorption from the gut by polyvalent cations. The vitamins and minerals are widely cation-containing Fe, Ca, or Mg supplements (HIV guidelines. clinicalinfo.hiv.gov. Available at https://clinicalinfo.hiv.gov/en/guidelines/adult-and-adolescent-arv/interactions-between-integrase-inhibitors-and-non-nucleoside.) The publication bias did not provide the P-value and the results of the Funnel plots should be clearly presented.

Finally, there was to have an impression that this study did not result in any new observations from reading the Discussion section.

Minor comments,

  1. The "title", please provides key information about the main objective or question the review addresses (e.g. the population(s) and intervention(s) the review addresses).
  2. In the Introduction, please describe the aim of the study using the " PICO"
  3. (If the purpose is to evaluate the effects of interventions, use the Population, Intervention, Comparator, Outcome (PICO) framework or one of its variants, to state the comparisons that will be made.)
  4. In the Introduction, please revise the phrase " tenofovir + lamivudine". (page 2, Line 41)
  5. In the Introduction, " Like wise, such interaction may modulate the efficacy of the drug via alteration of pharmacokinetic or pharmacodynamic mechanisms" The statement is unclear. (P2 L58-59)
  6. In the Methods, please add the reference for Systematic Reviews and Meta-Analyses (PRISMA) guidelines. (P2 L86)
  7. Provide registration information for the review, including register name and registration number, or state that the review was not registered. (P2 L87)
  8. In the Methods, please describe clearly" Specify the inclusion criteria and exclusion criteria" including study characteristics (year of publication, country, type of in-vitro studies.), language, the phase of clinical tries, case reports, and papulations, and ARV. (P3 L97)
  9. In the Methods, please confirm the authors had contact with the 28 included studies to get the original data. (P3 L97)
  10. In the Methods, please clearly define the primary outcome assessments (data items) and add the references. (P3 L97)
  11. In the Results, please provide a brief summary of the characteristics of study design and risk of bias among studies, estimate and its precision, and measures of statistical heterogeneity.
  12. In the Results, please label the number of references in Table 1.
  13. In the Results, please confirm the study " Roberts et al., 2011" in Table 1.
  14. In the Results, please confirm the study, reference No. 15.
  15. In the Discussion, the authors did not well review the supplements interactions with antiretroviral therapy (ART) and individually discuss the pharmacokinetic outcomes of the interaction between supplements and them. (P3 L221~232)
  16. Except for the limitations of the study, the authors should discuss any limitations of the evidence included in the review. Also, please discuss any limitations of the review processes used, and comment on the potential impact of each limitation.

Author Response

Major comments,

The authors conducted a systematic review and meta-analysis to clarify the pharmacokinetic outcomes of the interaction between supplements and antiretroviral agents (ARV). Basically, the pharmacokinetic outcomes were varied between the people were living with HIV (PLWH) and healthy subjects (HIV-negative).

Authors’ reply: We appreciate the reviewer’s suggestions. Responses to reviewer 2 comments are highlighted in green.

  1. The authors should be clearly defined the primary outcome in the study design to stratify the two groups (PLWH and healthy subjects) and the classification of ARV. Hence, the authors need to carefully present the data in PLWH on ARV. Therefore, I suggest the authors perform the subgroup analysis. For primary outcomes, the pooled proportions between the PLWH and healthy subjects, as well as ARV were calculated, the extremities sample size was observed between the two groups. And the test heterogeneity was obviously found in Figures 2, 3, and 5. The authors should well discuss in the main text and in the limitation of the study.

            - We did the stratification as your recommendation in Figures 3, 5, 6.

            We performed the subgroup analysis according to ARV regimens, modified figures and added the statements as followings. 

Figure 2 shows the meta-analysis of the pharmacokinetic effects of food on the AUCinf of ARV. There were no significant changes in AUCinf of ARV between fasted and fed conditions (mean difference 9.8, 95% CI -14.35 – 33.95, p = 0.43), with a substantial heterogeneity of 75%.

The overall effect of food on Cmax of NRTI group shows no clinical significance between fasted and fed conditions (mean difference -0.56, 95% CI -2.31 – 1.20, p = 0.53), with the heterogeneity value of 84%. Interestingly, there was a significant increase in Cmax of PI group during fed state (mean difference 334.98 (159.55 – 510.40, p = 0.0002) as shown in Figure 3. Figure 4 indicates the overall effect of food on C24 which was not significant (mean difference -0.01, 95% CI -0.67 – 0.65, p = 0.98).

The results of the subgroup meta-analysis of pharmacokinetic changes of tmax of ARV depending upon food or supplement administration are shown in figure 5. The overall effect in NRTI group indicates a significantly higher tmax value under fed condition (mean difference 0.68, 95% CI 0.53 – 0.82, p < 0.00001), showing the heterogeneity of 83%. Similarly, we found a significant increase in tmax in fed state compared to fasted condition in PI receiving groups (mean difference 1.16, 95% CI 0.98 – 1.34, p < 0.00001) (Figure 5). For both NRTI and PI groups, none of these shows any clinical significant changes in t1/2 (Figure 6). We observed no statistical heterogeneity on t1/2 of both NRTI and PI regimens.

[Please see the revised figures in the response-to-reviewer 2 PDF file or in the manuscript]

Figure 3. Forest plot showing the mean difference in Cmax of ARV under fasted and fed states by ARV regimens

[Please see the revised figures in the response-to-reviewer 2 PDF file or in the manuscript]

Figure 5. Forest plot showing the mean difference in tmax of ARV under fasted and fed states by ARV regimens

[Please see the revised figures in the response-to-reviewer 2 PDF file or in the manuscript]

Figure 6. Forest plot showing the mean difference in t1/2 of ARV under fasted and fed states by ARV regimens

  1. Given all the currently available protease inhibitors (PIs) are metabolized by the cytochrome P450 (CYP) enzyme system. All are inhibitors of CYP3A4, ranging from weak inhibition for saquinavir to very potent inhibition for ritonavir. (Malaty LI, Kuper JJ. Drug interactions of HIV protease inhibitors. Drug Saf. 1999 Feb;20(2):147-69.) Nucleoside reverse transcriptase inhibitors (NRTIs) do not undergo hepatic transformation through the CYP450. NRTIs would be less have the interaction with supplements. Non-nucleoside reverse transcriptase inhibitors (NNRTIs) are metabolized in the liver by CYP3A isoenzymes. Their pharmacokinetic outcomes were varied and complex. Also, the main concern regarding interactions with integrase strand transfer inhibitors (INSTIs) is the potential for decreased absorption from the gut by polyvalent cations. The vitamins and minerals are widely cation-containing Fe, Ca, or Mg supplements (HIV guidelines. Clinicalinfo.hiv.gov. Available at https://clinicalinfo.hiv.gov/en/guidelines/adult-and-adolescent-arv/interactions-between-integrase-inhibitors-and-non-nucleoside.)

            - Thank you for your comment. We add this information to the Discussion.

  1. The publication bias did not provide the P-value and the results of the Funnel plots should be clearly presented.

            - The evaluation of the funnel plot was performed by visual inspection without further Egger’s or Begg’s test since the plot was derived from only 5 studies. The Cochrane Library recommends those tests when there are 10 studies or higher to prove valid data. We add this sentence in the Result to further clarify this.

            “Further statistical test for the funnel plot asymmetry was not performed since the forest plot that had the largest number of included studies included only five studies [45].”

Reference

45        Cochrane handbook for systematic reviews of interventions. Recommendations on testing for funnel plot asymmetry  [cited 16 January 2022]. Available from: https://handbook-5-1.cochrane.org/chapter_10/10_4_3_1_recommendations_on_testing_for_funnel_plot_asymmetry.htm.

  1. Finally, there was to have an impression that this study did not result in any new observations from reading the Discussion section.  

                  - We do apologize if we did not clearly explain. We add this sentence to the paragraph.

            “This study found new information on the potential impact of supplement use on ARV pharmacokinetics. The study highlighted decreased absorption of NRTIs such as abacavir, emtricitabine and PIs such as ritonavir and darunavir after co-administration with food. The increased tmax and plasma concentration of some ARV such as darunavir due to ARV-food interaction can suggest nutrition monitoring in relation to HIV and ARV treatment.”

Minor comments,

1. The "title", please provides key information about the main objective or question the review addresses (e.g. the population(s) and intervention(s) the review addresses).

            - We appreciate the reviewer’s comment. Since our study focused on the variations in ARV pharmacokinetic parameters such as AUC, Cmax, we changed our title into “Pharmacokinetic outcomes of the interactions of antiretroviral agents with food and supplements: a systematic review and meta-analysis”.

2. In the Introduction, please describe the aim of the study using the " PICO". (If the purpose is to evaluate the effects of interventions, use the Population, Intervention, Comparator, Outcome (PICO) framework or one of its variants, to state the comparisons that will be made.)

            - Thank you for your comment. We change our aim according to your comment. A sentence below is added to the Introduction.

“This systematic review and meta-analysis aimed to investigate the effect of food, dietary supplements, or nutrients on pharmacokinetic outcomes of ARV by comparing the pharmacokinetic parameters in either PLWH or healthy people with and without supplements.”

3. In the Introduction, please revise the phrase " tenofovir + lamivudine". (page 2, Line 41)

            - We revised “tenofovir + lamivudine” into “tenofovir/lamivudine”.

4. In the Introduction, " Likewise, such interaction may modulate the efficacy of the drug via alteration of pharmacokinetic or pharmacodynamic mechanisms" The statement is unclear. (P2 L58-59)

            - We modified the sentence as “Likewise, food-drug interaction may influence the therapeutic efficacy of the drug by changing pharmacokinetic processes such as absorption, distribution, metabolism, and excretion, or pharmacodynamic physiological effects of the drug.”

5. In the Methods, please add the reference for Systematic Reviews and Meta-Analyses (PRISMA) guidelines. (P2 L86)

            - Reference number 16 for Systematic Reviews and Meta-analyses (PRISMA) guidelines was added in the method section.

Reference

16        Page MJ, McKenzie JE, Bossuyt PM, Boutron I, Hoffmann TC, Mulrow CD, et al. The PRISMA 2020 statement: an updated guideline for reporting systematic reviews. BMJ. 2021;372:n71.

6. Provide registration information for the review, including register name and registration number, or state that the review was not registered. (P2 L87)

            - We did not register our systematic review. Thus, we described in the method section as “The full protocol of this systematic review and meta-analysis was not registered.”

7. In the Methods, please describe clearly" Specify the inclusion criteria and exclusion criteria" including study characteristics (year of publication, country, type of in-vitro studies.), language, the phase of clinical tries, case reports, and papulations, and ARV. (P3 L97)

            - We described the inclusion and exclusion criteria as “All articles of any language, year published and country which reported nutrient or food supplement and ARV interactions were included to retrieve relevant studies. Only studies that reported pharmacokinetic outcomes such as AUC, Cmax, tmax, and t1/2 were included in the meta-analysis. Review articles, book chapters, conference abstracts, posters, in vitro studies, and animal studies were excluded. The search results were exported to a citation manager (Endnote 20.1., Clarivate Analytics, New York, USA). Titles and abstracts were thoroughly screened, and eligible studies were independently selected by the first two authors for inclusion in this systematic review. Specific characteristics for inclusion were studies of adult healthy people or PLWH on ARV which discussed changes in ARV levels, concerned adverse events, or treatment failure directly resulting from the food-drug interaction (Phase I-IV clinical trials).”

8. In the Methods, please confirm the authors had contact with the 28 included studies to get the original data. (P3 L97)

            - We did not contact any authors of the included studies.

9. In the Methods, please clearly define the primary outcome assessments (data items) and add the references. (P3 L97)

            - We added this sentence to the Methods.

            “The primary outcomes that we aimed to collect were AUC, Cmax, and Tmax which are parameters used for primarily assessed the food effect on drug pharmacokinetics [17].”

10. In the Results, please provide a brief summary of the characteristics of study design and risk of bias among studies, estimate and its precision, and measures of statistical heterogeneity.

            - We add paragraphs to the Result.

            “Thirteen articles of them were good quality and 11 articles were fair (Supplementary material 2).”

            “In general, the heterogeneity was high and the 95% confidence intervals of the pooled data were close to the most weighted included studies whose intervals were narrow.”

11. In the Results, please label the number of references in Table 1.

            - In table 1, the reference numbers were labeled.

12. In the Results, please confirm the study " Roberts et al., 2011" in Table 1.

            - We respond to this comment as we understood. If this response is not answering the comment, please let us know.

            We included case reports also in our systematic review. Roberts et al reported the subtherapeutic levels of raltegravir after concomitant calcium administration in an HIV-infected man.

13. In the Results, please confirm the study, reference No. 15.

            - We respond to this comment as we understood. If this response is not answering the comment, please let us know.

            Reference No.15 (Roberts et al) suggested caution is required when integrase inhibitors such as raltegravir are used in combinations with supplements or polyvalent cation containing antacids.

14. In the Discussion, the authors did not well review the supplements interactions with antiretroviral therapy (ART) and individually discuss the pharmacokinetic outcomes of the interaction between supplements and them. (P3 L221~232)

            - We appreciate the reviewer’s comments. Accordingly, we categorized the discussion into the impacts of food, vitamin and mineral, and dietary supplements on the pharmacokinetics of ARV as follows.    

The different classes of ARV have varied pharmacokinetic metabolisms. PIs are extensively metabolized by the cytochrome P (CYP) 450 enzymes in the liver and small intestine [46]. Thus, ingestion of some foods or juices that have an inhibitory effect on CYP450 may increase concentrations of PIs. Kupferschmidt et al [24] reported an increase in the bioavailability of saquinavir bioavailability after an intake of grapefruit juice, which is widely known as a CYP3A4 inhibitor [47]. In addition, a previous study reported an increase in AUC of atazanavir, lopinavir/ritonavir, nelfinavir, and saquinavir when ingested with food [48]. Under the fed condition, there was a long tmax of amprenavir from 1 to 4 hours compared with the fasting state [28]. The tmax of darunavir under fasted conditions was approximately 1.5 hours whereas it was increased to 3-4 hours after administration with food including standard breakfast, croissant with coffee, and high-fat breakfast. Similarly, the median tmax of 2 hours of ritonavir without food extended to 4 or 5 hours when food was administered concurrently [29]. Regarding the changes in plasma concentration of PI such as darunavir, Sekar et al [29] reported the increased Cmax by about 30% under fed condition compared with the fasted state. Another class of ARV, NNRTIs such as efavirenz and nevirapine are also metabolized by several liver CYP isoenzymes [49]. The administration of efavirenz with high fat or high caloric meal was associated with the increased mean AUC by 28% and mean Cmax by 29%, respectively compared to fasted conditions [50]. Most NRTIs are degraded by liver enzymes from the purine or pyrimidine nucleoside salvage pathway, depending upon the NRTI analogs [51]. Since they are not extensively metabolized by CYP450 [51], they have less interaction with food or other drugs.

The impacts of vitamin and mineral supplements on ARVs were mostly investi-gated in IIs [15, 35-37] and some PIs [34, 38]. Co-administration of 1,200 mg calcium carbonate and 324 mg ferrous fumarate with dolutegravir during fasted conditions de-creased in AUC0-∞, Cmax, and C24 of dolutegravir by respective 37% to 39% and 54% to 57% due to chelation with polyvalent cations [36]. However, coadministration of these divalent cations and dolutegravir under fed conditions counteracted such the interaction [36]. Moreover, Roberts et al [15] reported subtherapeutic raltegravir levels when co-administered with 1,000 mg calcium carbonate. Nonetheless, information on the time interval between raltegravir and calcium intake or whether they were administered under fed or fasted conditions were not reported. Given that calcium supplement is necessary for the prevention of osteoporosis, especially in PLWH [54], future studies investigating the magnitude of such interaction between raltegravir and calcium supplement under fasted and fed conditions should be conducted. If the fed condition can counteract the interaction, calcium supplements may be coadministered with raltegravir. Concerning the impacts of vitamin and mineral supplements on PIs, while the divalent calcium ion did not significantly alter nelfinavir and its active metabolite M8 concentrations [38], co-administration with vitamin C significantly decreased steady-state indinavir concentrations [15].  This could be explained by the inductive property of vitamin C on CYP isoenzymes observed in animal studies [55-59]. However, another study reported no significant effect of vitamin C on CYP3A4, the primary enzyme used to metabolize indinavir activity [60]. Based on the contradictory effects of vitamin C on CYP3A4, further studies are needed to confirm the results.

Concerning the impacts of various dietary supplements, different classes of ARVs were investigated, mostly the PIs. Piscitelli et al [39] reported that garlic supplement significantly decreased the extent of saquinavir absorption, as indicated by the decrease in AUC0-8, Ctrough, and Cmax. No definite underlying mechanism could be drawn from this study, but the authors proposed that it could be due to the induction of CYP450, or P-glycoprotein (P-gp) produced by long-term use of garlic. Nonetheless, evidence has shown that the effects of garlic on CYP3A4 are controversial as some in vitro studies reported that garlic had an inhibitory effect on CYP3A4 [61]. While other studies showed no significant effect of garlic on CYP3A4 [62, 63]. Given that garlic supplements might have a negative effect on saquinavir exposure, garlic supplementation should be avoided in patients receiving treatment with saquinavir.

Quercetin, an inhibitor of various CYP450 including CYP3A4 as well as P-gp, could theoretically be used as a booster of saquinavir levels [64, 65]. However, DiCenzo et al [40] did not find significant effects of quercetin on saquinavir concentrations. The nonsignificant effect of quercetin could partially be explained by the intersubject and intrasubject variability. Given that only ten subjects were included in the study by DiCenzo et al [40], further studies are required to confirm such results.

Ginkgo biloba, another widely used dietary supplement, has been shown to induce CYP3A activity [66, 67]. However, Robertson et al [41] reported no significant effects of Ginkgo biloba extract on lopinavir and ritonavir pharmacokinetics. These could be explained by the inhibitory effect of co-administered ritonavir on CYP3A. Nevertheless, the impacts of Ginkgo biloba extract on un-boosted protease inhibitors were not investigated in that study, and hence, coadministration of Ginkgo biloba extract with un-boosted PI is not theoretically recommended. Similar to the effect of Ginkgo biloba, Panax ginseng has shown an inductive effect on CYP3A, which in turn may reduce PI concentrations [68]. However, the two-week administration of Panax ginseng did not alter lopinavir and ritonavir pharmacokinetics [42]. The same reason concerning the inhibitory effect of concurrently administered ritonavir could be applied here. Therefore, a similar recommendation on the use of Panax ginseng and un-boosted PI to that of Ginkgo biloba extract is proposed.

Inconclusive effects of β-carotene on CYP3A4 have been reported from in vitro studies, as one study found an inhibitory effect [69], while another reported an induc-tive property [70]. A clinical study investigating the impacts of β-carotene supplemen-tation at the dose of 25,000 IU twice daily on nelfinavir and its active metabolite M8, which is metabolized by respective CYP2C19 and CYP3A4, indicated no clinically sig-nificant effect [44]. This implies that PLWH receiving ARVs that are substrates of CYP2C19 or CYP3A4 may be able to use β-carotene as a dietary supplement. However, it should be noted that these findings were based only on 15 HIV-infected subjects. Further prospective studies investigating the impact of β-carotene on other CYP2C19 and CYP3A4 substrates may be warranted.

15. Except for the limitations of the study, the authors should discuss any limitations of the evidence included in the review. Also, please discuss any limitations of the review processes used, and comment on the potential impact of each limitation.

            - We appreciate the reviewer’s input. The additional details of the limitations of the study were mentioned.

            “This study has some limitations. The included studies did not report the component of fruit juices i.e., whether the juice contains pulps or additives. This information may be crucial since these compositions might not be pharmaceutically inert. Since most studies were conducted with different ARV drugs and different supplements, only limited studies were included for meta-analysis. Only food (meal) and ARV pharmacokinetic interaction was analyzed due to the limited number of studies with supplements and ARV pharmacokinetic interaction. Some studies from the qualitative analysis could not be included in the meta-analysis because of the diverse types of supplements among studies. The number of participants in each study is relatively small, and there may also have interpatient or intrapatient variability. Furthermore, most studies were conducted in different regions with different study designs. Moreover, our systematic review may not be generalizable to all nutrient or food supplements as we did not focus on some products e.g. probiotics or prebiotics which may have effects on ARV pharmacokinetics. Of note, the heterogeneity across studies may also affect the results. Thus, the quality of this systematic review may be influenced by those factors.”

Round 2

Reviewer 2 Report

In the Results, please confirm the study, reference No. 15. 

- We respond to this comment as we understood. If this response is not answering the comment, please let us know. 

Reference No.15 (Roberts et al) suggested caution is required when integrase inhibitors such as raltegravir are used in combinations with supplements or polyvalent cation containing antacids. 

ROBERTS, Jennie L., et al. Virologic Failure with a Raltegravir‐Containing Antiretroviral Regimen and Concomitant Calcium Administration. Pharmacotherapy: The Journal of Human Pharmacology and Drug Therapy, 2011, 31.10: 1042-1042.

Author Response

We edited the data of the reference no.15 according to the reviewer's response. Thank you very much for your kindness.